# Circadian Rhythm of NER and ATR Pathways

**DOI:** 10.3390/biom11050715

**Published:** 2021-05-11

**Authors:** Tae-Hong Kang

**Affiliations:** Department of Biomedical Sciences, Dong-A University, Busan 49315, Korea; thkang@dau.ac.kr; Tel.: +82-51-200-7261

**Keywords:** circadian clock, DNA damage response, ultraviolet radiation (UV), DNA repair, nucleotide excision repair (NER), ataxia-telangiectasia-mutated and Rad3-related (ATR), chronotherapy, cyclobutane pyrimidine dimer (CPD), 6-4 photoproduct (6-4 PP)

## Abstract

Genomic integrity is constantly insulted by solar ultraviolet (UV) radiation. Adaptative cellular mechanisms called DNA damage responses comprising DNA repair, cell cycle checkpoint, and apoptosis, are believed to be evolved to limit genomic instability according to the photoperiod during a day. As seen in many other key cellular metabolisms, genome surveillance mechanisms against genotoxic UV radiation are under the control of circadian clock systems, thereby exhibiting daily oscillations in their catalytic activities. Indeed, it has been demonstrated that nucleotide excision repair (NER), the sole DNA repair mechanism correcting UV-induced DNA photolesions, and ataxia–telangiectasia-mutated and Rad3-related (ATR)-mediated cell cycle checkpoint kinase are subjected to the robust control of the circadian clock. The molecular foundation for the circadian rhythm of UV-induced DNA damage responses in mammalian cells will be discussed.

## 1. Introduction

Everlasting day-night cycles and the accompanying environmental oscillations have placed evolutionary pressure on most organisms on Earth. To accommodate these recurring environmental changes, organisms have been equipped with an internal oscillating system to keep track of light-dark cycles, the “circadian clock system.” The evolutionarily conserved feature of the circadian clock across most living organisms strongly suggests that it must confer selective advantages [1].

Rhythmic oscillations in physiology and behavior with a period close to 24 h are called circadian rhythms, which are generated by the cell-autonomous molecular clock operating virtually in every single cell [2]. The mammalian molecular clock system is a conserved transcriptional and translational autoregulatory feedback loop through the concerted action of the heterodimeric transcription activator’s CLOCK-BMAL1 complex, which induces the transcription of target genes that contain E-box elements (CACGTG) in their promoter and/or enhancer regions. The gene expression of transcriptional repressors cryptochrome (CRY) and period (PER) are also induced by CLOCK-BMAL1 activity, thereby creating self-sustainable 24 h rhythms in gene expression. [3]. This transcriptional circuitry generates daily oscillations of output genes for the temporal regulation of cell-specific physiology, which ultimately maintains the metabolic homeostasis of an organism [4]. Consequently, the chronic disruption of circadian regulation as a result of shiftwork or other lifestyle factors predisposes to the onset of numerous chronic diseases such as cardiovascular diseases, cancer, and aging [5].

Ultraviolet (UV) radiation from sunlight consists of three different wavelengths termed UVA (315–400 nm), UVB (280–315 nm), and UVC (100–280 nm). The one with shorter wavelengths generates more DNA lesions efficiently. Indeed, UVB and UVC readily respond to DNA to produce genotoxic photoproducts such as cyclobutane pyrimidine dimers (CPD) and pyrimidine–pyrimidone (6-4) photoproducts (6-4PP) [6]. Because these photolesions can interfere with pivotal DNA metabolisms, including replication and transcription, if not repaired promptly by an error-free repair system, they can accumulate mutations and eventually provoke genomic instability, a hallmark of both cancer and aging [7,8]. Indeed, increased recreational exposure to sunlight has significantly contributed to the surge in recent skin cancer cases [9], and prolonged exposure to solar UV also can cause a plethora of acute and chronic symptoms, including sunburn erythema, skin thickening, and photoaging [10].

Nucleotide excision repair (NER) is the sole and error-free DNA repair mechanism capable of correcting UV-induced DNA photolesions in placental mammals [11]. Based on the genomic position of the photolesion, two distinct NER subpathways comprising the global genome repair (GGR) and the transcription-coupled repair (TCR) can be operating optionally. The GGR, in which a DNA lesion not undergoing transcription is repaired, is activated by the DNA damage sensor proteins, xeroderma pigmentosum group C (XPC) and DDB complex (DDB1 and DDB2), whereas the TCR, in which damage to the transcribing strand of DNA is repaired, is initiated by the stalling of RNA polymerase II through its ability to sense the blocking of transcript elongation [12].

NER is exceptional among the DNA repair systems in its ability to eliminate the widest class of structurally unrelated DNA lesions, including UV-photolesions, chemical-DNA bulky adducts, reactive oxygen species induced base alterations, and intrastrand crosslinks [13]. The NER system consists of more than 30 proteins, including seven XP proteins from XPA to XPG, and two Cockayne syndrome proteins, CSA and CSB [14].

In addition to activating the NER mechanism, UV-photolesions trigger the ataxia–telangiectasia-mutated and Rad3-related (ATR) checkpoint pathway to transiently arrest the cell cycle to allow sufficient time for NER [15]. To stimulate the kinase activity of ATR, a common DNA structure consisting at least partly of single-stranded DNA covered with replication protein A (RPA) is a prerequisite. This structure can be generated during the NER process as an intermediate and also can serve as a platform for the recruitment of ATR-interacting protein (ATRIP), which facilitates ATR recruitment to the damaged site.

Consequently, the NER and the ATR pathways are expected to be interconnected intimately to each other to cope with UV-photolesions on genomic DNA. The recent findings on the circadian rhythm of NER and the ATR pathways and possible crosstalk between the two systems will be discussed to shed light on the role of the circadian clock system in UV-evoked DNA damage responses and ultimately to translate it into clinical application for human health.

## 2. Circadian Oscillation of XPA and the NER Activity

Circadian oscillations of metabolic processes are pervasive and play key roles in ensuring homeostatic balance with the environment to coordinate virtually every aspect of the cellular event, physiology, and behavior [16]. Likewise, the activity of genome surveillance mechanisms against the inevitable attack from solar UV is also highly dependent on the fitness of molecular clock circuitry. Indeed, the gene expression of the key genes catalyzing the mechanism is under the control of the circadian clock and exhibits high-amplitude daily oscillations [17].

The efficiency of NER is essential for the maintenance of genome integrity against UV irradiation in placental mammals, including humans, because it represents the sole system capable of neutralizing the two major UV-photolesions (CPD and 6-4PP) on genomic DNA. Mutations in genes associated with the NER mechanism, consequently, cause a wide range of cutaneous symptoms, from mild solar sensitivity to severe skin cancers [18].

XPA is an essential component of both subpathways for NER, and its transcriptional and post-transcriptional regulation may have significant effects on cellular repair and survival following exposure to UV and UV-mimetic agents [19]. Functionally, XPA mediates the damage verification, and this serves as a confirmation signal for dual incision by XPF and XPG endonucleases [13]. Consequently, the steady-state level of XPA directly dictates the capacity of DNA lesion removal by NER [20].

A decade ago, the circadian rhythm of XPA and NER activity was observed in mammalian tissues. To gain insight into the mechanistic foundation underlying this phenomenon, Kang et al. analyzed the levels of key components consisting of the NER system and revealed that only the level of XPA was governed by the circadian clock, showing a daily oscillation with maximum and minimum levels closely correlating with the NER activity [21]. By demonstrating that XPA is a clock-controlled gene exhibiting robust circadian rhythm in its both transcript and protein levels, he provided for the first time compelling evidence that DNA repair in mammals is controlled by the circadian clock [22].

In line with this report, a study by Gaddameedhi et al., in which a mouse skin was analyzed for assessment of circadian rhythmicity of NER, also confirmed that the transcription of the XPA gene is regulated positively by the CLOCK-BMAL1 complex and negatively by CRY-PER complex [23]. The peak NER activity coincides with maximum XPA protein levels, indicating the importance of XPA in determining the NER capacity [21,23,24]. It is advantageous for organisms to have elevated levels of NER activity at the time of the day of maximal exposure to sunlight. It is noteworthy that circadian oscillation of XPA levels was not uniformly observed in all tissue types (e.g., absent in actively proliferating tissues such as testis [24] and tumor [25]) and may contribute to a range of NER thresholds among different tissues [26]. Coupled with circadian transcriptional regulation, XPA protein is subjected to constitutive ubiquitin-dependent degradation by the HERC2 ubiquitin ligase, independent of circadian oscillation, thus ensuring the prompt removal of XPA protein, contributing to a robust circadian oscillation of XPA protein and the subsequent NER activity (Figure 1A) [24].

With few exceptions for a protein to demonstrate robust circadian oscillation, the gene encoding it must first be transcribed with circadian rhythmicity, and secondly, the protein must have a relatively short lifetime; even if a gene is transcribed with circadian rhythm, it would not show high-amplitude oscillation if it is stable [27]. HERC2, a large HECT- and RCC-like domain-containing protein [28], is found to regulate NER activity by ubiquitinating and degrading XPA [20,29]. In addition to NER, HERC2 is implicated in regulating other DNA damage response pathways, such as homologous recombination repair through its effect on BRCA1 [30] and replication stress response through that on CLASPIN [31]. Frameshift mutations in HERC2 have been found in both gastric and colorectal carcinomas with microsatellite instability [32]. The HERC2 locus has also been associated with both cutaneous melanoma and uveal melanoma [33,34]. 

HERC2 is initially reported to function as a scaffold to recruit a RING finger E3 ligase 8 (RNF8) to the sites of double-strand breaks to aid in double-strand break repair [30]. In the study, while knockdown of either HERC2 or RNF8 sensitizes cells to ionizing radiation, a catalytically inactive HERC2 mutant complemented HERC2-depleted cells, but an active-site mutant of RNF8 failed to complement RNF8-depleted cells for resistance to ionizing radiation. Therefore, a conclusion has been drawn that RNF8 is the active E3 ligase within the HERC2-RNF8 complex, implying that the enzyme activity of HERC2 is dispensable. However, a series of follow-up studies established that HERC2 is an enzymatically active E3 ligase for the turnover of its targets, including BRCA1 [35], USP20 [36,37], and XPA [20].

ATR has been reported as a binding partner of XPA, and this interaction can substantially increase the NER activity and cell viability in response to UV damage [38]. To this end, ATR phosphorylates XPA at serine residue 196 (S196) [39], which results in the increased protein stability of XPA by attenuating HERC2-catalyzed XPA ubiquitination (Figure 1B) [29]. Specifically, the report demonstrates that upon UV irradiation ATR facilitates HERC2 dissociation from the XPA complex, which results in the accumulation of XPA proteins. However, this regulation is abrogated in S196A (a phospho-deficient form of XPA)-complemented XPA-deficient cells due to persistent association of HERC2 with this form of XPA, resulting in enhanced ubiquitination of S196A protein. Conversely, the S196D (a phospho-mimic form of XPA) substitution shows delayed degradation kinetics compared with the wild-type XPA due to the resistance of HERC2 association, resulting in reduced ubiquitination of S196D protein, hence enhanced NER capacity. Therefore, it is likely that any factor engaged in the activation of ATR kinase such as direct ATR activators (TOPBP1 [40] and ETAA1 [41]), mediators (CLASPIN [42] and TIMELESS [43]), and proximal regulators (PKA [44], NDR1 [45], and TTP [46]) would positively affect NER activity through either directly or indirectly modulating the ATR pathway.

Meanwhile, it has been reported that XPA is acetylated at lysines 63 and 67 by an unknown acetyltransferase, and this acetylation significantly reduced XPA function in NER by interfering with the XPA-RPA interaction [47]. It is also reported that SIRT1, a NAD(+)-dependent histone deacetylase, binds to XPA and prevents its acetylation or, if XPA is acetylated, SIRT1 deacetylates and thus activates it [48]. In support of this model, a recent study demonstrated that downregulation of SIRT1 significantly reduced both the repair rate of CPDs and the survival of UV-irradiated cells. This report raised the interesting possibility that the circadian rhythm of NER can be generated by the XPA acetylation/deacetylation rhythm because both SIRT1 [49,50] and the NAD(+)-synthesizing enzyme, NAMPT [51,52], exhibit circadian rhythms. Thus, it is reasonable that the circadian rhythms of NAD(+) and SIRT1 generate an acetylation–deacetylation cycle of XPA with circadian periodicity, resulting in a daily oscillation of NER activity. However, a recent study revealed that in mouse liver, less than 5% of XPA is acetylated at a given time of the day, which means that over 95% of XPA is active at all times, and activation of less than 5% of XPA is not expected to make a substantial contribution to the rate of NER [20]. In any event, the experiments on XPA acetylation and the NER rate both in mouse liver and in human cell lines yield results consistent with the conclusion that SIRT1 deacetylation of XPA does not contribute to the circadian rhythmicity of NER [20].

Apart from the UV-photolesion repair, the removal of a chemotherapeutic agent cisplatin-induced intrastrand crosslink is strongly dependent on the circadian clock due to the circadian oscillation of XPA expression [24]. Cisplatin, and its second- and third-generation derivatives, make two major DNA adducts, Pt-(GpG) and Pt-(GpTpG) [53]. Extensive data indicate that these intrastrand diadducts are the major cause of cytotoxicity and that NER is the primary repair system for these adducts [54]. The cellular response to genotoxic agents is dictated by pharmacodynamic factors, including the cell cycle phase, DNA repair capacity, apoptosis, and circadian clock. Hence, in designing a chemotherapy regimen, all these factors should be taken into account. Ultimately, the implications for the administration of DNA-damaging therapy by the guidance of the patient’s circadian clock, so-called “chrono-modulated chemotherapy,” represents a novel strategy of personalized cancer therapy for minimizing adverse side effects, thereby improving the cancer management and patient outcomes [55].

## 3. Cryptochrome Control of the ATR-Mediated Checkpoint Pathway

DNA damage-induced cell cycle checkpoints are signal transduction pathways activated following DNA damage and serve as a genome surveillance mechanism to coordinate multiple cellular pathways for ensuring genomic integrity [15]. The ATR (ATM and Rad3-related) and ataxia-telangiectasia-mutated (ATM)-mediated signaling pathways represent two major DNA damage-induced cell cycle checkpoints in mammals [56]. These pathways are composed of proteins in the four conceptual categories of DNA damage sensors, signal mediators, transducers, and downstream effector molecules [57]. While ATM is activated to orchestrate DNA double-strand break (DSB) repairs in response to DSBs, the ATR checkpoint pathway is mainly triggered by single-strand breaks and base modifications, including the damage generated by UV irradiation [58]. ATR is targeted to the sites of elongated replication protein A (RPA)-covered single-strand DNA. This event is mediated by interactions between RPA and the ATR interaction protein (ATRIP) [59]. Upon sensing DNA damage, ATR initiates a complex signaling cascade via phosphorylation of downstream protein substrates such as CHK1 whose activation leads to cell cycle arrest [46].

Apart from their canonical clock function, the mammalian clock factors have also been implicated in diverse noncanonical functions such as the DNA damage response (DDR), which includes cell cycle checkpoint [60,61] and DNA repair [62,63]. First, as the most pervasive transcription factor in differentiated cells, the CLOCK-BMAL1 complex affects DDR capacity by controlling the rate of transcription of genes involved in genotoxic responses [64,65]. Second, clock proteins sometimes directly interact with and activate DDR factors for maintaining genomic integrity. Therefore, failure in such intimate crosstalk between clock and DDR factors can result in genomic instability and tumorigenesis [66,67,68]. For instance, mice with a homozygous mutation of PER2 show deregulated cyclin D and c-MYC expression and spontaneous tissue hyperplasias and lymphomas [69]. Although transcriptional regulation by the circadian clock partially affects tumorigenesis, recent findings now suggest that each clock factor may play a specific role in DDR by participating as a key modulator. For instance, PER1 interacts with the ATM-CHK2 complex in response to DSBs [67].

The clock repressor cryptochromes (CRYs) are indispensable for molecular clockwork [70] and robustness to circadian timekeeping [71]. Naturally, animals lacking CRY1 have short periods, those lacking CRY2 have long periods, and animals lacking both CRYs (CRY^DKO^) are arrhythmic under constant darkness [72]. Intriguingly, in response to UV damage, CRY1, independently of its canonical clock function, was capable of regulating the ATR activity by forming a complex with TIMELESS (TIM) in a time-of-day-dependent manner (Figure 2).

The ATR activity measured by the level of the phosphorylation status of ATR substrates such as CHK1 and MCM2 exhibits circadian rhythm. Interestingly, clock-deficient CRY^DKO^ cells retained substantial ATR activity compared with clock-proficient wild-type cells, although the CRY1-modulated rhythmic ATR activity was abolished in CRY^DKO^ cells. The molecular foundation of this phenomenon is driven by the temporal interaction of CRY1 and TIM in the nucleus due to the cyclical expression of CRY1 protein. Depending on the level of nuclear CRY1, the differential ATR activity was observed, eventually generating a circadian oscillation of ATR activity. And when CRY1 was ectopically expressed in CRY^DKO^ cells, the rhythmic ATR activity was recurred. Importantly, the significantly altered ATR activity in mouse liver that was intraperitoneally injected with cisplatin at different circadian times was detected, which consequently affected the removal rate of cisplatin-DNA adducts from genomic DNA in the liver. Collectively, this study demonstrates the intimate interaction between the circadian clock and the ATR pathway during genotoxic stress in clock-ticking cells. In line with the story, CRY1 has recently been characterized as a protumorigenic factor that promotes most DNA repair mechanisms (NER, MMR, BER, HR, NHEJ) and cell survival through temporal transcriptional regulation, nominating CRY1 as a new therapeutic target [63].

## 4. Crosstalk between NER and ATR Pathways

Several studies have already demonstrated that UV-induced ATR activation is dependent on NER in the quiescent cells [73,74]. The activated ATR then directly facilitates NER by phosphorylating and activating XPA. The NER mechanism excises a ~30 nucleotide oligomer containing the damage [75]. The resulting gap is filled in by DNA polymerases, however, if this process is delayed or defective, this NER-intermediate structure can be further processed by exonuclease 1 (EXO1), a 5′ → 3′ exonuclease. The resulting extended ssDNA coated with RPA now can serve as a structural requirement for ATR–ATRIP recruitment and checkpoint activation [76]. Subsequently, ATR is ready to participate in the activation of NER by phosphorylating XPA (Figure 3).

The melanocortin 1 receptor (MC1R) regulates pigmentation, adaptive tanning, and melanoma resistance by activating adenylyl cyclase which accumulates intracellular cyclic AMP (cAMP) levels [77]. Upon UV exposure, MC1R-cAMP signaling promotes PKA-mediated phosphorylation of ATR at Ser435, a modification that enhances NER by facilitating recruitment of the XPA protein to sites of UV-induced DNA photolesions [44]. Similarly, an XPA-interacting kinase named NDR1 (nuclear-Dbf2-related) facilitates CPD removal by activating the ATR pathway [45]. The lesion recognition factors in the GGR subpathway, XPE, and XPC alternatively can activate the ATR pathway by recruiting ATR kinase to the sites of UV damage [78]. On the contrary, a protein phosphatase WIP1 has been shown to negatively regulate NER kinetics by dephosphorylating and inactivating XPA [79].

Importantly, a defective NER activity exclusively during the S phase of a majority of human melanoma cell lines is speculated as a result of decreased ATR signaling, which may constitute an unrecognized determinant in melanoma pathogenesis [80].

## 5. Concluding Remarks

In the last 20 years, most of the core factors dedicated to the NER mechanism have been elaborately described, shifting attention to the mechanisms that facilitate NER in a spatiotemporal context and to cooperative interactions between NER and other signaling pathways. While increased NER activity protects the genome against the accumulation of DNA lesions, thereby maintaining genome integrity, it might be beneficial to reduce the NER capacity in patients with cancer who are undergoing chemotherapy, because doing so might help ensure the efficient action of DNA damage-inducing drugs such as cisplatin. In that regard, we expect that the transient suppression of NER through the pharmacological manipulation of NER core factors or ATR pathways will synergize with DNA-damaging agents to optimize chemotherapeutic outcomes. A wide range of transcriptional and post-translational regulatory mechanisms of NER factors has been revealed recently, providing attractive targets to adjust the NER threshold [6]. Detailed mechanistic understanding of these regulatory pathways will be necessary to guide genetic and pharmacological manipulations for research and disease intervention.

Chronotherapeutics aim at treating illnesses according to the endogenous biological rhythms, which moderate pharmacokinetics and pharmacodynamics of a certain drug [81]. Given that NER, the primary repair mechanism for removal of chemotherapeutic agent cisplatin, has a circadian rhythm in healthy tissues but is relatively arrhythmic in cancer tissues, we can individualize the most favorable time for drug administration, which will be when the NER activity is the highest in normal tissues of a patient. Pharmacological modulation of core clock genes is an alternative approach in cancer therapy [82]. The integration of circadian biology into cancer research will offer new options for making cancer treatment more effective.

## Figures and Tables

**Figure 1 biomolecules-11-00715-f001:**
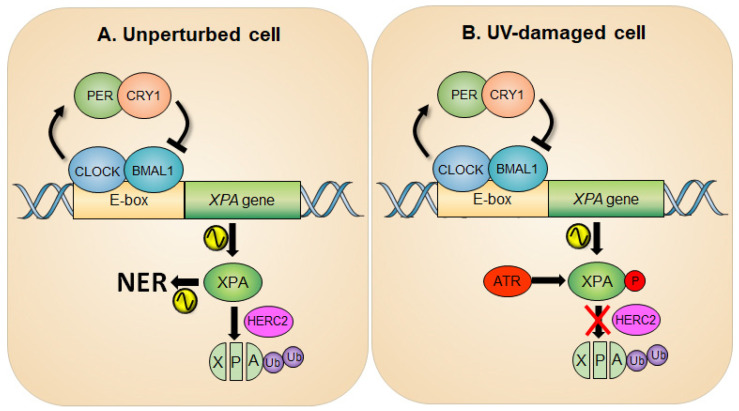
Circadian rhythm of XPA and NER activity in an unperturbed (**A**) and UV-damaged (**B**) cell. (**A**) The promoter of the XPA gene possesses E-box sequences for CLOCK-BMAL1-driven transcription of the gene, thereby generating a circadian rhythm of XPA and NER activity. The half-life of XPA is short due to the action of an E3 ubiquitin ligase HERC2, which adds ubiquitin to XPA and degrades it through the proteasome pathway, making the amplitude of circadian NER activity robust. (**B**) Upon UV damage, the ATR checkpoint pathway is activated and then phosphorylates XPA at Ser196. This results in an increase in XPA protein stability by attenuating HERC2-induced XPA ubiquitination and consequently upregulates the NER activity.

**Figure 2 biomolecules-11-00715-f002:**
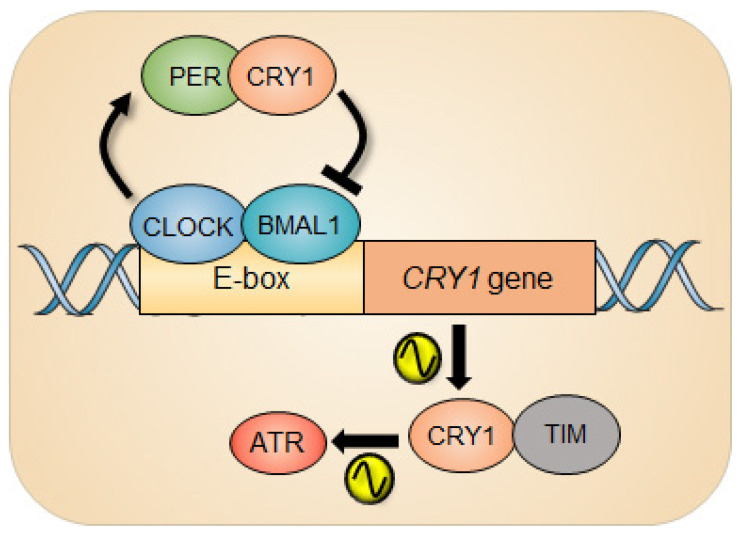
Circadian rhythm of ATR activity. The circadian repressor CRY1 is a clock-controlled gene whose gene expression is under the control of the circadian clock. In response to UV damage, CRY1 can regulate the ATR activity by interacting with TIMELESS (TIM) as a function of time, which generates circadian oscillation of ATR activity.

**Figure 3 biomolecules-11-00715-f003:**
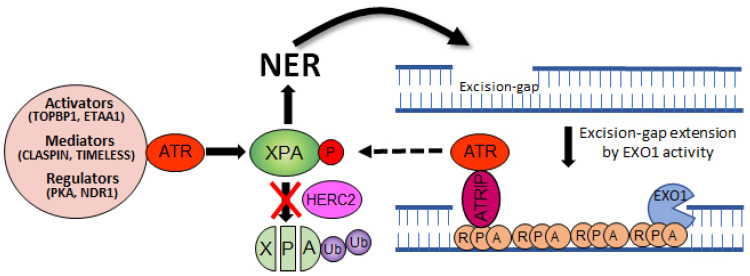
Crosstalk between NER and ATR pathways. The NER activity excises a ~30 nucleotide oligomer containing the damage. The resulting excision gap is filled in by DNA polymerases, however, if this process is delayed or defective, the gap can be processed by EXO1 nuclease. Subsequently, a single-strand binding protein RPA covers the extended gap to limit DNA double-strand break. This intermediate structure is favored by the ATR-ATRIP complex and followed by XPA phosphorylation and NER activation, making a positive feedback loop between the two pathways.

## Data Availability

Not applicable.

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
