# Peer review of "Circadian Rhythm of NER and ATR Pathways"

_biomolecules, 2021, doi:10.3390/biom11050715_

Round 1
Reviewer 1 Report
This is a very interesting review about the remarkable control of the nucleotide excision repair (NER), the sole DNA repair mechanism correcting UV–induced DNA lesions, and ATR (ataxia–telangiectasia mutated and Rad3–related)-mediated cell cycle check-point kinase by the circadian clock. The author gives strong literature evidence for the administration of cancer therapy orchestrated by the circadian clock in order to maximize the efficacy of the treatment and minimize the side effects.
Minor comments
- Standardize nomenclature of proteins and genes. For example: Lines 266-267 - “For instance, Per1 interacts with the ATM–CHK2 complex in response to DSBs [67].”
This Per1 probably is a transcription fator, that is a protein? Should it be in all capital letters?
- Please rephrase: lines 22-24 - “To accommodate these recurring environmental changes, the organisms have developed an internal oscillating system to keep track of light–dark cycles, the ‘circadian clock system’.”
The way it’s written it’s mesleading and induces the reader to assume that the clock appeared in response to environmental pressure...
- Place Figure 2 after its citation in the text
- Please correct gramar:
Lines 81-83 - “Indeed, the gene expression of the key genes catalyzing the mechanism are under the control of the circadian clock and are exhibiting a high -amplitude daily oscillations [17].” “Indeed, the gene expression of the key genes catalyzing the mechanism is under the control of the circadian clock and exhibits a high -amplitude daily oscillations [17].”
Lines 115,116 - “thus ensuring the prompt removal of XPA protein, contributing a robust circadian oscillation of XPA protein and the subsequent NER activity (Figure 1A) [24].” “thus ensuring the prompt removal of XPA protein, contributing to a robust circadian oscillation of XPA protein and the subsequent NER activity (Figure 1A) [24]”
Line 165 - “Therefore, a conclusion has drawn that RNF8 is the active”
“Therefore, a conclusion has been drawn that RNF8 is the active”
Line 224-225 – “the ATR checkpoint pathway mainly is triggered by”
“the ATR checkpoint pathway is mainly triggered by”
Lines 287-288 – “In line with the story, CRY1 has recently characterized as a pro-tumorigenic factor that promotes”
“In line with the story, CRY1 has recently been characterized as a pro-tumorigenic factor that promotes”
Line 292 – “Interestingly, several studies have already been demonstrated that UV-induced ATR”
“Interestingly, several studies have already demonstrated that UV-induced ATR”
Lines 309-310 – “On the contrary, a protein phosphatase WIP1 has been shown that it negatively regulates NER”
“On the contrary, a protein phosphatase WIP1 has been shown to negatively regulate NER”
Line 355 – “but relatively arrhythmic in câncer tissues”
“but is relatively arrhythmic in câncer tissues”
- Line 159 - Please delete comma between HERC2 AND is initially reported to function as a scaffold
6)Lines 182-183 - What is the meaning of et al in the sentence: “(CLASPIN [42], TIMELESS [43], et al), and proximal regulators (PKA [44], NDR1 [45], TTP [46], et al).”
7) Lines 312-313 – Should'nt it be “human melanoma cell lines”? “Importantly, a defective NER activity exclusively during the S phase of a majority of human melanoma cells is speculated”
Author Response
I am submitting a revised version of the manuscript “Circadian rhythm of NER and ATR pathways”.
I am very grateful for the criticisms and the suggestions from the reviewer. All the comments made by the reviewers have been addressed and clarified as detailed in the point-by-point reply, and I have modified the manuscript accordingly.
I hope that the overall revision makes for an acceptable manuscript, and I hope to hear about this soon.
With best regards,
Tae-Hong Kang, Ph.D.

Reviewer 2 Report
This work is an excellent review of circadian rhythm of nucleotide excision repair (NER) and ataxia-telangiectasia mutated and rad3-related (ATR) pathways.
The manuscript is written very nicely and clearly. I recommend publication of this paper in Biomolecules in the present form.
Author Response
Thanks for your positive and fruitful comments on my manuscript.
Reviewer 3 Report
T Hong described that NER and ATR pathways are regulated by circadian rhythm. The level of XPA is the sole mediator of NER that governed by circadian rhythm. ATR-mediated phosphorylated XPA is resistant to degradation by ubiquitin ligase HERC2. The checkpoint mediator ATR is also regulated by circadian rhythm. XPA and ATR make a positive feedback loop. T Hong introduced “chronochemotherapy” to maximize cancer treatment efficacy and minimize side effects.
- In the introduction section, there seems little description about cell cycle checkpoint governed by ATR when compared to the description of NER. Therefore, the author should describe the details of the ATR-mediated checkpoint.
- I am a little bit confused with the function of HERC2. Is HERC2 regulated by circadian rhythm? If not, why the author describes the HERC2 function? As described in Fig.1B, ATR-mediated phosphorylated XPA is resistant to degradation by HERC2. Given that ATR is regulated by circadian rhythm, the whole story of HERC2 should be described in the setting of the ATR section because this review focuses on circadian rhythm.
- The story of acetylation of XPA governed by SIRT1 and NAMPT is a little bit hard to understand. Could the author use a figure to explain it?
- The story of “chronochemotherapy” is little bit hard to understand. In case chemotherapy is done in the daytime, both efficacy of cancer treatment and side effects are expected to minimized while in case it is done in at night, both factors are expected to maximized. It is hard to imagine the case of “chronochemotherapy” with maximized efficacy of cancer therapy and minimized side effects.
Author Response

(The authors gave the same response as above.)
